# Group-wise Verifiable Distributed Computing for Machine Learning under Adversarial Attacks

## Abstract

Distributed computing has been a promising solution in machine learning to accelerate the training procedure on large-scale dataset by utilizing multiple workers in parallel. However, there remain two major issues that still need to be addressed: i) adversarial attacks from malicious workers, and ii) the effect of slow workers known as stragglers. In this paper, we tackle both problems simultaneously by proposing Group-wise Verifiable Coded Computing (GVCC), which leverages coding techniques and group-wise verification to provide robustness to adversarial attacks and resiliency to straggler effects in distributed computing. The key idea of GVCC is to verify a group of computation results from workers at a time, while providing resilience to stragglers through encoding tasks assigned to workers with Group-wise Verifiable Codes. Experimental results show that GVCC outperforms the existing methods in terms of overall processing time and verification time for executing matrix multiplication, which is a key computational component in machine learning and deep learning.

## 1 Introduction

Recently, machine learning and big data analysis have achieved huge success in various areas such as computer vision, natural language processing, and reinforcement learning, etc. Since they usually demand a massive amount of computation on a large dataset, there has been increasing interest in distributed systems, where one node is used as a master and the others are used as workers.

One possible option is distributed computing (Dalcín et al., 2005; 2011), where the workers compute partial computation task received from the master. In distributed computing, the master divides and distributes tasks (which require far small memory than the original task) to workers, and they compute the assigned tasks and send results back to a master. Distributed computing can be utilized to compute matrix multiplication in machine learning, the most important and frequent computation block. In a distributed setting, however, there exist two foremost considerations to embed distributed computing in machine learning applications, i) stragglers and ii) adversarial workers.

Stragglers are workers that return their computation results much slower than others. It has been reported that stragglers can be a serious bottleneck to performing large-scale computation tasks (Dean & Barroso, 2013; Huang et al., 2017; Tandon et al., 2017). To handle straggler effects, coded computing was first suggested in Lee et al. (2018). In coded computing, a master encodes a computation task with a coding technique while retaining redundancy in task allocation. Due to the redundancy arisen from the coding technique, a master does not need all results of tasks to achieve the final output and can ignore stragglers. This approach has been applied to various computation tasks, especially on matrix multiplication (Dutta et al., 2016; Yu et al., 2017; Park et al., 2018; Reisizadeh et al., 2019; Dutta et al., 2019; Yu et al., 2020).

Moreover, some of the workers could be adversarial workers, which send perturbed results to the master to contaminate or degrade the performance of neural networks. Many studies (Biggio et al., 2012; Blanchard et al., 2017; El Mhamdi et al., 2018; Sohn et al., 2020; Bagdasaryan et al., 2020; Wang et al., 2020) demonstrate that adversarial workers slow down the overall training process and

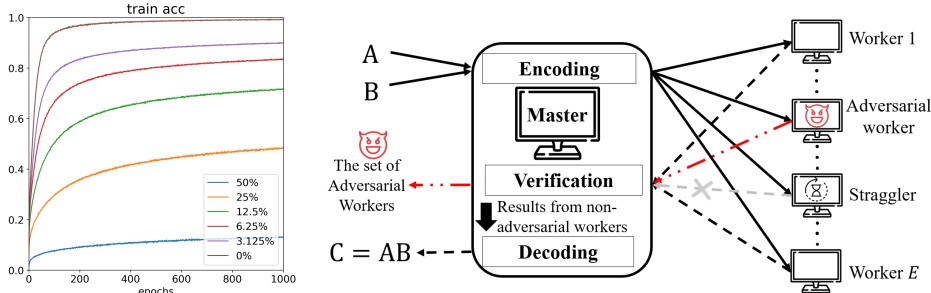

(a) Training curve of neural networks under adversarial attacks.

(b) Distributed computing for a matrix multiplication $\mathbf{C} = \mathbf{AB}$ under adversarial workers and stragglers.

**Figure 1:** Training curve of neural networks under adversarial workers and system model for distributed computing

severely degrade the performance of a neural network. In this paper, we mainly consider attacks from adversarial workers (i.e. Byzantine workers) in distributed computing, where they try to contaminate the final computation product by returning wrong computation results for the assigned task.

To demonstrate the effect of adversarial workers in distributed computing, we trained a neural network under adversarial attack. Fig 1(a) shows the training curve of neural networks on CIFAR-100. 32 workers were used for training and adversarial workers send random values with the same magnitude as the original results during training. As we can see in the figure, the training accuracy of a neural network degraded severely when there exist adversarial workers. The training accuracy of a neural network converged to 0.99 after 560 epochs when there is no adversarial worker. However, with only 3.125% of adversarial workers, the training accuracy of the neural network was only 0.89 after 1000 epochs.[1]

To be robust against adversarial attacks, a master can check whether the computation results from workers are correct or not to figure out adversarial workers. Based on this idea, there have been some recent studies that suggest an encoding scheme for assigning computation tasks to workers to tolerate wrong computation results when obtaining the final computation product, and to identify adversarial workers from the returned computation results (Yu et al., 2019; Soleymani et al., 2021; Hong et al., 2022). Another line of research, which we call verifiable computing, has focused on identifying adversarial workers with a fast verification procedure. In verifiable computing, a master verifies the correctness of computation results by using a verification key (Freivalds, 1977; Tang et al., 2022).

In this paper, we focus on designing a fast and robust distributed computing system for matrix multiplication. We use both a coding approach and verifiable computing to combat adversarial attacks from malicious workers and straggler effects. Our system model is depicted in Fig 1(b), where a master encodes the computation tasks, distributes them to the workers, and decodes the final product after receiving and verifying the computation results from the workers.

To be specific, we suggest a novel encoding scheme that enables verifying a large number of computation results at a time, and decoding the final computation product from a part of computation results from the workers to mitigate straggler effects as well. We experiment with various settings and demonstrate that our proposed scheme can identify adversarial workers and obtain the final computation product much faster than existing methods. The main contributions of this paper are as follows.

- We propose **Group-wise Verifiable Coded Computing (GVCC)**, handling both straggler effects and adversarial attacks in distributed computing for matrix multiplication tasks. To be more specific, we suggest group-wise verifiable codes (GVC) for encoding the computation tasks of workers and provide a suitable group-wise verification algorithm.

---

[1]We provide more explanation and detailed experiment settings in Appendix A.

We first provide a process of group-wise verification trial (GVT), then propose a two-stage verification algorithm based on GVT. We also provide modified verification algorithm that can be utilized under straggler effects.

- We demonstrate the performance of GVCC via experiments in terms of overall processing time and verification time for executing matrix multiplication tasks in distributed computing under adversarial attacks and straggler effects. According to experimental results, GVCC speeds up overall processing time and verification time up to $\times 1.11$ and $\times 3.34$ compare to that of the existing verifiable coded computing scheme in Tang et al. (2022), and speeds up up to $\times 2.46$ and $\times 174$ than group testing based adversarial attack identification scheme for coded matrix multiplication in Hong et al. (2022), respectively.

**Notations.** For $a, b \in \mathbb{Z}$, $[a : b]$ denotes $\{a, a + 1, \ldots, b\}$.

## 2 PROBLEM SETTING

We consider a distributed computing scenario for matrix multiplication task with one master and $E$ workers $\{W_i\}_{i=1}^E$. A master wants to perform a matrix multiplication $\mathbf{C} = \mathbf{AB}$ on input matrices $\mathbf{A} \in \mathbb{F}_q^{a \times b}$ and $\mathbf{B} \in \mathbb{F}_q^{b \times c}$ for a sufficiently large finite field $\mathbb{F}_q$. The master generates encoded matrices using $\mathbf{A}$ and $\mathbf{B}$, then distributes them to workers to allocate tasks. To be specific, the master encodes $\tilde{\mathbf{A}}_i$, $\tilde{\mathbf{B}}_i$ and sends them to $W_i$ for $i \in [1 : E]$. After receiving $\tilde{\mathbf{A}}_i$ and $\tilde{\mathbf{B}}_i$, each worker $W_i$ computes the allocated task

We assume that some of the workers could be adversarial workers or stragglers. We list our assumptions on adversarial workers and stragglers as follows:

- There exist $l$ adversarial workers $W_{\mathcal{L}}$, where $\mathcal{L}$ denotes a set of adversarial workers for $\mathcal{L} \subset [1 : E]$ and $|\mathcal{L}| = l$.
- Stragglers occur randomly depending on the network condition and currently available computing resource at each worker.
- Adversarial workers have knowledge about input matrices $\mathbf{A}$ and $\mathbf{B}$, and all the protocol of distributed computing including encoding, task allocation, and decoding, only except the random key $\mathbf{r}$ that is used for verification.[2]
- A master cannot be aware of the number of adversarial workers in advance.
- Adversarial workers cannot collude with each other, which implies that they cannot access the information about assigned tasks of other workers.

To improve the performance of distributed computing systems under adversarial attacks and stragglers, we aim to reduce verification time and overall processing time of distributed matrix multiplication.

## 3 BACKGROUND: VERIFIABLE COMPUTING AND CODED COMPUTING

In this section, we first introduce two methods that are leveraged in GVCC: i) verifiable computing that enables identifying adversarial workers and ii) coded computing that mitigates the effects of stragglers.

### 3.1 VERIFIABLE COMPUTING FOR ADVERSARIAL ATTACK

The authors of Freivalds (1977) proposed a verification method (Freivalds' method) that uses a random key to verify the computation results. Freivalds' method for matrix multiplication can be summarized in three steps as follows.

**Computation assignment**: A master assigns a matrix multiplication task $\mathbf{C} = \mathbf{AB}$ for $\mathbf{A} \in \mathbb{F}_q^{a \times b}$ and $\mathbf{B} \in \mathbb{F}_q^{b \times c}$ to an available worker. The worker executes the assigned task, and returns its result $\mathbf{C}'$ to the master.

---

[2]Protecting the random key from the adversarial workers can be realized by randomly generating $\mathbf{r}$ at the master, after receiving results from the workers.

**Verification key generation**: The master generates a random key $\mathbf{r} \in \mathbb{F}_q^{1 \times a}$ and computes a verification key $\mathbf{s}_1 = \mathbf{rA}$.

**Correctness check**: The master computes $\mathbf{s}_2 = \mathbf{s}_1 \mathbf{B} = \mathbf{rAB} = \mathbf{rC}$ and $\mathbf{s}_2' = \mathbf{rC}'$ from the received result. By comparing $\mathbf{s}_2$ and $\mathbf{s}_2'$, the master can verify whether $\mathbf{C}'$ is correct or not.

It has been known that the master can identify adversarial workers using this verification process with probability $1 - \frac{1}{q}$ in the finite field $\mathbb{F}_q$ (Sahraei & Avestimehr, 2019). In this verification procedure, the master computes $ab + bc + ca$ scalar products while the original task $\mathbf{C} = \mathbf{AB}$ requires $abc$ scalar products.

## 3.2 CODED COMPUTING FOR STRAGGLERS

In distributed computing, several methods have been proposed to mitigate straggler effects. One traditional method is to replicate a computational task and allocate it to multiple workers (Wang et al., 2014). However, this replication-based method is inefficient as it requires $K$ additional workers to tolerate a single straggler, when the computational task is divided into $K$ smaller tasks. To handle straggler effects more efficiently, coded computing was suggested in Lee et al. (2018). By using a coding theoretic approach for allocation of tasks to workers, the master can retrieve the final product after receiving enough number of workers. We now define an important metric that has been used in many coded computing researches.

**Recovery threshold** $R$: The minimum number of computation results of tasks from non-adversarial workers that the master requires to obtain the final product for the worst-case scenario.[3]

In coded computing, the master can decode the final output from $R$ out of $E$ results and it is able to tolerate up to $E - R$ stragglers. Therefore, coded computing requires $S$ additional workers for handling $S$ stragglers, while the replication-based method requires $KS$ additional workers for $S$ stragglers.

## 4 GROUP-WISE VERIFIABLE CODED COMPUTING

In this section, we provide GVC and propose an algorithm for group-wise verification. We also suggest a modified algorithm for GVCC under straggler effects.

### 4.1 ENCODING OF GROUP-WISE VERIFIABLE CODES (GVC)

We now explain the encoding procedure of GVC that facilitates group-wise verification at the master. In GVC, the master divides input matrices $\mathbf{A}$ and $\mathbf{B}$ into sub-matrices of equal size $\mathbf{A}_w \in \mathbb{F}_q^{\frac{a}{m} \times b}$ for $w \in [1 : m]$, and $\mathbf{B}_z \in \mathbb{F}_q^{b \times \frac{c}{n}}$ for $z \in [1 : n]$. Then input matrices and their product $\mathbf{C} = \mathbf{AB}$ are given by

$$\mathbf{A} = \begin{bmatrix} \mathbf{A}_1 \\ \vdots \\ \mathbf{A}_m \end{bmatrix}, \quad \mathbf{B} = [\ \mathbf{B}_1 \ \cdots \ \mathbf{B}_n\ ], \quad \mathbf{C} = \begin{bmatrix} \mathbf{A}_1 \mathbf{B}_1 & \cdots & \mathbf{A}_1 \mathbf{B}_n \\ \vdots & \ddots & \vdots \\ \mathbf{A}_m \mathbf{B}_1 & \cdots & \mathbf{A}_m \mathbf{B}_n \end{bmatrix}. \quad (1)$$

To generate the encoded matrices, the master uses encoding functions $\mathbf{p_A}$ and $\mathbf{p_B}$ constructed by using sub-blocks of $\mathbf{A}$ and $\mathbf{B}$ as coefficients, which are given by

$$\mathbf{p_A}(x) = \Sigma_{i=1}^m \mathbf{A}_i x^{i-1}, \quad \mathbf{p_B}(x) = \Sigma_{j=1}^n \mathbf{B}_j f^{j-1}(x), \quad (2)$$

where $x$ represents the variable of encoding functions $\mathbf{p_A}$ and $\mathbf{p_B}$. $f(x)$ is the basic building component for constructing $\mathbf{p_B}$ and we now introduce the conditions on $f(x)$ that enable group-wise verification of GVC.[4]

---

[3]It should be noted that previous works have included stragglers or adversarial workers in the recovery threshold and in this case, the recovery threshold of GVC can be expressed as $R' = R + S + l$.

[4]The condition for $f(x)$ was also used in squeezed polynomial codes (Hong et al., 2020), in order to reduce communication load in distributed computing.

**Condition I** : The polynomial function $f(x)$ used in the encoding function $\mathbf{p_B}(x)$ should satisfy the following conditions.

i) $f(x)$ is an $m$th order polynomial function.

ii) $f(x)$ has at least $\frac{E}{m}$ sets of $m$ evaluation points $\alpha_{1,1}, \alpha_{1,2}, ..., \alpha_{\frac{E}{m},m}$ that satisfy $f(\alpha_{t,1}) = f(\alpha_{t,2}) = \cdots = f(\alpha_{t,m}) = w_t$ for $t \in \left[1 : \frac{E}{m}\right]$.

cUsing the encoding functions and evaluation points satisfying Condition I, the master generates the encoded matrices $\tilde{\mathbf{A}}_{t,u}$ and $\tilde{\mathbf{B}}_{t,u}$ for $t \in \left[1 : \frac{E}{m}\right]$ and $u \in [1 : m]$ as

$$\tilde{\mathbf{A}}_{t,u} = \mathbf{p_A}(\alpha_{t,u}) = \Sigma_{i=1}^{m}\mathbf{A}_i\alpha_{t,u}^{i-1},$$

$$\tilde{\mathbf{B}}_{t,u} = \mathbf{p_B}(\alpha_{t,u}) = \Sigma_{j=1}^{n}\mathbf{B}_j f^{j-1}(\alpha_{t,u}) = \Sigma_{j=1}^{n}\mathbf{B}_j w_t^{j-1} = \tilde{\mathbf{B}}_{t,*},$$

$$(3)$$

where $w_t = f(\alpha_{t,u})$ for $u \in [1 : m]$, which are the values of encoding functions $\mathbf{p_A}$ and $\mathbf{p_B}$ at the evaluation points $x = \alpha_{t,u}$.

**Remark 1**. (Quantization and embedding into the finite field) In this paper, we assume that input matrices are defined on a finite field $\mathbb{F}_q$, while the dataset for machine learning and polynomial function $f(x)$ used for encoding are based on the real field $\mathbb{R}$. To express real field data and function in the finite field, we quantize matrices $\mathbf{A}, \mathbf{B}$, and $f(x)$ using $v$ bits, and embed them into a finite field $\mathbb{F}_q$ of integers at modulo a prime $q$. Quantization of real numbers and embedding into the finite field have been used in several previous works (Yu et al., 2017; Ji et al., 2021; Tang et al., 2022). To analyze the effect of quantization in training of the deep neural network, we provide an experiment in Appendix B.

**Remark 2.** (Polynomials for GVC) GVC can be constructed using any polynomial satisfying Condition I. For instance, Chebyshev polynomial, which has been used in several coded computing studies (Fahim & Cadambe, 2021; Hong et al., 2021), can be used as $f(x)$ by using quantization as stated in **Remark 1**. We also provide the proof in Appendix C.

**Remark 3.** (Reduced Computational overhead for encoding) In GVC, since it is guaranteed $\tilde{\mathbf{B}}_{t,u} = \tilde{\mathbf{B}}_{t,*}, \forall u \in [1 : m]$ for $t \in \left[1 : \frac{E}{m}\right]$, the master encodes $(1 + \frac{1}{m})E$ matrices to assign $E$ tasks, instead of encoding $2E$ matrices as in the conventional methods. Thus, the computational overhead for encoding can be lowered by GVC.

## 4.2 TASK ASSIGNMENT AND COMPUTING AT WORKERS

To assign the tasks to workers, the master randomly rearranges workers into $\frac{E}{m}$ groups of same size, which can be denoted as $W_{t,u}$ for $t \in \left[1 : \frac{E}{m}\right]$ and $u \in [1 : m]$, and distributes $\tilde{\mathbf{A}}_{t,u}, \tilde{\mathbf{B}}_{t,u}$ to $W_{t,u}$.

Each worker $W_{t,u}$ computes the assigned task $\tilde{\mathbf{C}}_{t,u} = \tilde{\mathbf{A}}_{t,u}\tilde{\mathbf{B}}_{t,u}$, which corresponds to the value of the evaluation function $\mathbf{p_C}(x)$ at the evaluation point $x = \alpha_{t,u}$, where $\mathbf{p_C}(x)$ is given by

$$\mathbf{p_C}(x) = \mathbf{p_A}(x) \times \mathbf{p_B}(x) = \Sigma_{i=1}^{m}\Sigma_{j=1}^{n}\mathbf{A}_i\mathbf{B}_j x^{i-1}f^{j-1}(x).$$

$$(4)$$

After each worker finishes the assigned task, it returns its computation result to the master.

## 4.3 GROUP-WISE VERIFICATION FOR GVC

After receiving the results from workers, the master begins a verification process. We first explain a group-wise verification process and suggest two-stage verification algorithm based on this process.

### 4.3.1 A GROUP-WISE VERIFICATION TRIAL (GVT)

Let us denote workers with finished computation among $\{W_{t,u}\}_{u=1}^{m}$ by $\mathcal{T}_t$ for $t \in [1 : \frac{E}{m}]$. By GVC, a master can verify results from group $\mathcal{T}_t$ in a single trial. In conventional verifiable computing approach, a master verifies the computation result $\tilde{\mathbf{C}}_i'$ for $i \in [1 : E]$ individually. However, if the tasks are encoded by GVC, it is ensured that $\Sigma_{\mathcal{T}_t}\tilde{\mathbf{C}}_{t,u} = \Sigma_{\mathcal{T}_t}\tilde{\mathbf{A}}_{t,u}\tilde{\mathbf{B}}_{t,u} = \Sigma_{\mathcal{T}_t}\tilde{\mathbf{A}}_{t,u}\tilde{\mathbf{B}}_{t,*} = \Sigma_{\mathcal{T}_t}\tilde{\mathbf{A}}_{t,u} \times \tilde{\mathbf{B}}_{t,*}$ for a group of workers $\mathcal{T}_t$ among $\{W_{t,u}\}_{u=1}^{m}$ where $|\mathcal{T}_t| = k$ $(k \leq m)$. Therefore,

the master can verify $k$ computation results $\tilde{\mathbf{A}}_{t,u}\tilde{\mathbf{B}}_{t,u}$ in a group by verifying $\Sigma_{\mathcal{T}_t}\tilde{\mathbf{A}}_{t,u}\times\tilde{\mathbf{B}}_{t,*}$ instead. On the basis of this fact, we suggest a process for group-wise verification trial (GVT).

**Verification key generation**: For each trial of group-wise verification, a master generates a random key $\bar{\mathbf{r}}_t \in \mathbb{F}^{1\times\frac{a}{m}}$ and computes a verification key $\bar{\mathbf{s}}_{1,t} = \bar{\mathbf{r}}_t\Sigma_{\mathcal{T}_t}\tilde{\mathbf{A}}_{t,u}$.

**Group-wise Correctness check**: The master computes $\bar{\mathbf{s}}_{2,t} = \frac{1}{k}\bar{\mathbf{s}}_{1,t}\Sigma_{\mathcal{T}_t}\tilde{\mathbf{B}}_{t,u} = \bar{\mathbf{s}}_{1,t}\times\tilde{\mathbf{B}}_{t,*} = \bar{\mathbf{r}}_t\Sigma_{\mathcal{T}_t}\tilde{\mathbf{A}}_{t,u}\times\tilde{\mathbf{B}}_{t,*} = \bar{\mathbf{r}}_t\Sigma_{\mathcal{T}_t}(\tilde{\mathbf{A}}_{t,u}\tilde{\mathbf{B}}_{t,u}) = \bar{\mathbf{r}}_t\Sigma_{\mathcal{T}_t}\tilde{\mathbf{C}}_{t,u}$ and $\bar{\mathbf{s}}'_{2,t} = \bar{\mathbf{r}}_t\Sigma_{\mathcal{T}_t}\tilde{\mathbf{C}}'_{t,u}$ from the received computation results. If $\bar{\mathbf{s}}_{2,t} \neq \bar{\mathbf{s}}'_{2,t}$, then the master marks the output of verification trial as **Positive**, which implies there exist more than one adversarial worker in the tested group $\mathcal{T}_t$, and marks as **Negative** otherwise.[5]

### 4.3.2 Two-stage verification algorithm

We now provide a two-stage verification algorithm of GVC, given by Algorithm 1.[6]

---

**Algorithm 1:** Group-wise Verifiable Coded Computing

---

1  **Two-stage group-wise verification**: After receiving all computation results from workers, the master begins group-wise verification.
2   **for** $t \in [1 : \frac{E}{m}]$ **do**
3     The master verifies $m$ workers using GVT in Section 4.3.1, where $\mathcal{T}_t = \{W_{t,u}\}_{u=1}^m$.
4     **if** *Group-wise verification result for $\mathcal{T}_t$ is negative* **then**
5         Put all workers in $\mathcal{T}_t$ to a set of non-adversarial workers $\mathcal{T}^n$.       // Stage 1
6     **else**
7         Put all workers in $\mathcal{T}_t$ to a set of possible adversarial workers $\mathcal{T}^p$.
8     **if** $|\mathcal{T}^n| \geq R$ **then**
9         break;
10   **if** $|\mathcal{T}^n| < R$ **then**
11     The master verifies each worker in $\mathcal{T}^p$ using GVT for $k = 1$, and put non-adversarial workers whose verification result is negative to $\mathcal{T}^n$ until $|\mathcal{T}^n| = R$.     // Stage 2

---

**Two-stage group-wise verification**: After receiving all computation results of tasks from workers, the master starts group-wise verification based on GVT. However, one thing we should take into account is that GVC can verify up to $m$ results at once. Thus, we propose a two-stage group-wise verification algorithm adjusted to the parameter $m$.

- **Stage 1:** The master groups $\{W_{t,u}\}_{u=1}^m$ as $\mathcal{T}_t$ for $t \in [1 : \frac{E}{m}]$. The master performs a group-wise verification on $\mathcal{T}_t$ using GVT.

- **Stage 2:** The master verifies each computation result in $\mathcal{T}_t$ if group-wise verification result for $\mathcal{T}_t$ is positive, until it achieves $R$ non-adversarial workers to obtain $\mathbf{C}$.

### 4.4 Decoding at the master

After verifying $R$ non-adversarial computation results, the master starts decoding. As we can see in equation 1 and equation 4, $\mathbf{A}_i\mathbf{B}_j$ for $i \in [1 : m]$, $j \in [1 : n]$ can be retrieved from $\mathbf{p}_\mathbf{C}(x)$ to achieve $\mathbf{C}$. Since $\tilde{\mathbf{C}}_{t,u} = \mathbf{p}_\mathbf{C}(\alpha_{t,u})$ and $\mathbf{p}_\mathbf{C}(x)$ is a polynomial of degree $mn-1$, it can be interpolated from $mn$ distinct value of $\tilde{\mathbf{C}}_{t,u}$. Hence, the master needs any fastest $mn$ non-adversarial computation results to decode the final output $\mathbf{C}$. Thus, the recovery threshold of GVC is $R = mn$.

---

[5]It should be noted that group-wise verification trial has the same accuracy with the individual verification in Freivalds (1977). We demonstrate this by experiment under various adversarial attacks in Appendix D.

[6]We do not consider straggler effects here for simplicity, but provide a modified verification algorithm of GVC that considers straggler effects in Section 4.5.

Decoding can be done by the inversion of the coefficient matrix in

$$
\begin{bmatrix} \tilde{\mathbf{C}}_1 \\ \vdots \\ \tilde{\mathbf{C}}_{mn} \end{bmatrix} = \left( \begin{bmatrix} {\beta_1}^0 & \dots & {\beta_1}^{m-1} & f(\beta_1) & \dots & {\beta_1}^{m-1} f(\beta_1)^{n-1} \\ \vdots & & \ddots & & & \vdots \\ {\beta_{mn}}^0 & \dots & {\beta_{mn}}^{m-1} & f(\beta_{mn}) & \dots & {\beta_{mn}}^{m-1} f(\beta_{mn})^{n-1} \end{bmatrix} \otimes I_{\frac{a}{m} \times \frac{a}{m}} \right) \times \begin{bmatrix} \mathbf{A}_1 \mathbf{B}_1 \\ \vdots \\ \mathbf{A}_m \mathbf{B}_n \end{bmatrix},
$$
(5)

where $\tilde{\mathbf{C}}_k$ and $\beta_k$ for $k \in [1 : mn]$ denote the fastest $mn$ computation results and the evaluation points of them, respectively and $\otimes$ denotes Kronecker product. It can be also achieved by interpolation (Kedlaya & Umans, 2011) of $\mathbf{p_C}(x)$ and extracting $\mathbf{A}_i \mathbf{B}_j$ from $\mathbf{p_C}(x)$ by repeated division of $f(x)$. We provide more detailed decoding process in Appendix E.

**Remark 4.** (Optimal Recovery threshold) It has been known in Yu et al. (2017) that the optimal recovery threshold for coded matrix multiplication is $mn$ when $\mathbf{A}$ and $\mathbf{B}$ are divided into $m$ and $n$ partitions in a row-wise and a column-wise manner, respectively. Thus *GVC also achieves the optimal recovery threshold*.

**Remark 5.** (Upper bound of tolerating adversarial attacks) Since GVCC requires $mn$ computation results to obtain the final product, GVCC can tolerate up to $l_{upper} = E - S - mn$ adversarial workers, where $S$ denotes the number of stragglers.

## 4.5 MODIFIED VERIFICATION ALGORITHM OF GVCC UNDER STRAGGLER EFFECTS

We now provide a modified algorithm that can be used under straggler effects.

**Two-stage group-wise verification in modified GVCC**:

- **Stage 1**: The master groups $\{W_{t,u}\}_{u=1}^m$ as $\mathcal{T}_t$ for $t \in [1 : \frac{E}{m}]$ as in GVCC. The master establishes a certain threshold $z$. The master starts a group-wise verification for the computation results of all groups received up to that time when the master receives $z$ computation results from workers.

- **Stage 2**: The master first individually verifies the computation results that arrived late, and verifies each computation result in $\mathcal{T}_t$ if group-wise verification result for $\mathcal{T}_t$ is positive, until it finds $R$ non-adversarial workers to obtain $\mathbf{C}$.

By setting a proper threshold $z$, the modified verification algorithm can efficiently deal with straggler effects. However, it may lose some of the benefits of group-wise verification as a trade-off because the master verifies a smaller group of workers in a single trial. The impact on overall processing time will be analyzed in Section 6 via experiment.

## 5 RELATED WORKS

In distributed computing, there have been several studies on handling adversarial attacks and straggler effects simultaneously. First, to have tolerance over adversarial attacks and provide resiliency to straggler effects, Lagrange coded computing (LCC) has been suggested in Yu et al. (2019). In LCC, a master requires $2K$ additional non-adversarial workers to be robust against $K$ adversarial workers. In addition, a list-decoding approach has been applied to distributed matrix multiplication in Subramaniam et al. (2019), and it reduces the additional cost of tolerating adversarial attacks by a factor of two asymptotically compared to LCC. However, the computational complexity to perform list-decoding increases quadratically with the number of workers.

Recently, verifiable computing has been introduced to coded computing schemes. The authors of Tang et al. (2022) have suggested AVCC to counter straggler and adversarial workers. In AVCC, a master only requires $K$ additional computation results from workers if there exist $K$ adversarial workers, because it can verify each computation result using a verification key. However, since the master can verify only a single computation result at a time, the verification process can be a bottleneck for overall processing if the master utilizes numerous workers in the system.

In addition, there have been several studies that are based on a group-wise identification to find adversarial workers, while mitigating straggler effects using coded matrix multiplication. First,

group testing algorithm has been applied to coded computing in Solanki et al. (2019). However, since it has a constraint on the minimum testable size for group testing, the number of test trials can be significantly large. To mitigate this limitation, hierarchical group testing (HGT) has been proposed in Hong et al. (2022). However, HGT imposes high computational complexity on a master since it requires matrix inversion or Reed-Solomon decoding to identify adversarial workers, whereas verifiable computing requires matrix-vector multiplications to verify them.

# 6 EXPERIMENTAL RESULTS

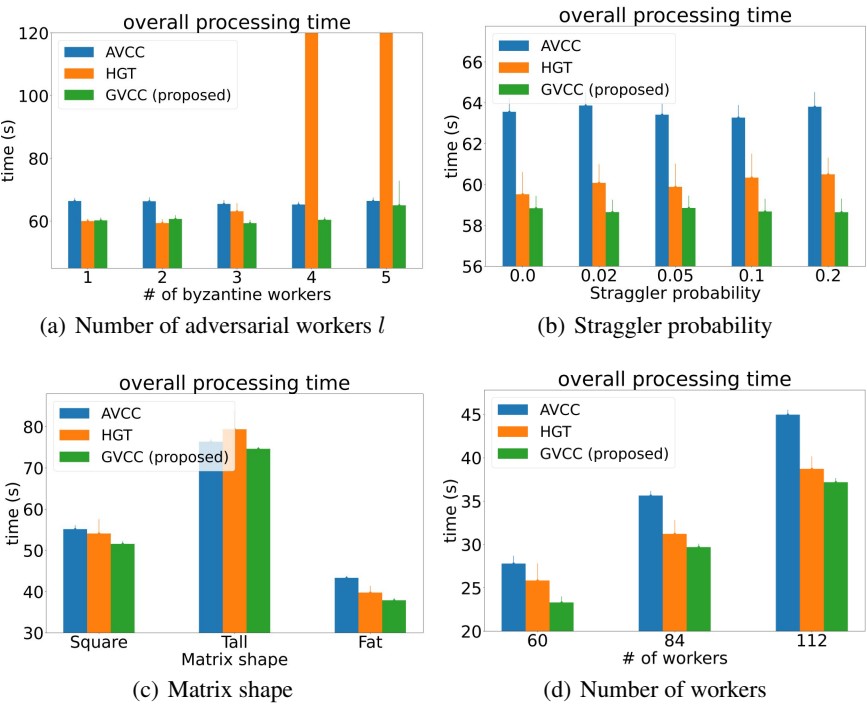

**Figure 2:** Overall processing time of AVCC, HGT, and GVCC (proposed).

In this section, we provide the performance of GVCC and modified GVCC in terms of overall processing time, and compare with the two existing schemes, AVCC (Tang et al., 2022) and HGT (Hong et al., 2022). We proceed distributed matrix multiplication in a cluster of Amazon EC2 cloud, and use one t2.2xlarge node as a master and use t2.micro nodes as workers. Implementation is done by MPI4py (Dalcín et al., 2005) and matrices are randomly generated by NumPy. We set the verification time limit as 100 seconds and if the verification process takes more than 100 seconds, we terminate the experiment and mark it as a failure. We run four experiments and evaluate the overall processing time. Each experiment is repeated 20 times and the result is depicted in Fig. 2. We also list stage-wise processing time (encoding (Enc), task assignment (SA), computation & return (CR), verification (Ver), decoding (Dec), and overall processing times) of experiments in Appendix F.1.

In Fig. 2(a), we compare overall processing time with different number of adversarial workers $l$, when $\mathbf{A} \in \mathbb{F}_q^{3500 \times 6300}, \mathbf{B} \in \mathbb{F}_q^{6300 \times 3600}, m = 5, n = 9$, and $E = 60$. GVCC speeds up overall processing time compared to AVCC for all cases, and also shows faster overall processing than HGT when $l \geq 3$. As we have claimed in Remark 3, the computational overhead for encoding the tasks can lowered by GVC, which results in reduced encoding and overall processing time compared to AVCC. Furthermore, for the verification time of HGT increases significantly with the number of workers, the verification and overall processing time of GVCC is much lower than HGT when $l \geq 3$. To be specific, GVCC speeds up verification time up to $\times 3.34$ than AVCC, and $\times 174$ than HGT, and HGT even fails to proceed verification process in the given time limit (100s) when $l = 5$.

In Fig. 2(b), we show the overall processing time with stragglers in distributed computing systems, where $\mathbf{A} \in \mathbb{F}_q^{3500 \times 6300}, \mathbf{B} \in \mathbb{F}_q^{6300 \times 3600}, m = 5, n = 9, E = 60$, and $l = 3$. To

simulate straggler effects in distributed computing, we randomly pick stragglers with probabilities $p = 0, 0.02, 0.05, 0.1$, and $0.2$. Stragglers are forced to run background thread which slows down the execution of assigned tasks. In this experiment, we use the modified algorithm in Section 4.5 for GVCC ($z = R = 63$) and compare with AVCC and HGT. Since verification is done asynchronously for the received computation results in GVCC, verification time is added to computation & return time. As we can see in the Fig 2(b), GVCC outperforms AVCC and HGT for all cases. Thus, it is claimed that GVCC can efficiently handle straggler effects while leveraging a group-wise verification approach. We also provide additional experiments with a modified algorithm on another parameter setting in Appendix F.2.

In Fig. 2(c), we compare overall processing time with different shape of input matrices with three parameter settings: i) Square ($\mathbf{A} \in \mathbb{F}_q^{3600 \times 3600}$, $\mathbf{B} \in \mathbb{F}_q^{3600 \times 3600}$), ii) Tall ($\mathbf{A} \in \mathbb{F}_q^{4500 \times 1500}$, $\mathbf{B} \in \mathbb{F}_q^{1500 \times 4500}$), iii) Fat ($\mathbf{A} \in \mathbb{F}_q^{2700 \times 7000}$, $\mathbf{B} \in \mathbb{F}_q^{7000 \times 2700}$), where $m = 5, n = 9, l = 3$, and $E = 60$. GVC speeds up overall processing time compared to AVCC and HGT, regardless of matrix shapes.

The performance of GVCC with the different numbers of workers is depicted in Fig. 2(d). We run experiments to multiply two input matrices $\mathbf{A} \in \mathbb{F}_q^{2100 \times 4000}$ and $\mathbf{B} \in \mathbb{F}_q^{4000 \times 3861}$, with three parameter setting $(E, m, n)$: i) $(60, 5, 9)$, ii) $(84, 6, 11)$, iii) $(112, 7, 13)$, where $l = 3$. GVCC achieves the lowest overall processing time in every setting. To be specific, GVCC speeds up verification time up to $\times 3.5$ than AVCC and $\times 57.29$ than HGT. Furthermore, GVCC speeds up overall processing time up to $\times 1.20$ than AVCC and $\times 1.10$ than HGT.

| Verification time (ms) | Group-wise verifiable size m | | | Matrix Size N | | |
|---|---|---|---|---|---|---|
| | 5 | 6 | 7 | 900 | 1800 | 3600 |
| HGT | 2750 | 1920 | 2440 | 2348 | 9554 | 42163 |
| AVCC | 140 | 170 | 210 | 16 | 45 | 163 |
| GVC | 48 | 48 | 46 | 5.6 | 17 | 62 |

**Table 1:** Verification time of GVC, AVCC, and HGT with different matrix size and $m$.

Furthermore, it should be noted that GVC shows more benefit than AVCC as $N$ or $m$ gets bigger. In specific, our methods have a verification complexity of $\mathbf{O}(\frac{R}{m}N^2)$ while AVCC has $\mathbf{O}(RN^2)$. To show the performance of GVC with different $m$ and $N$, we indicate the verification time of GVC and the other schemes in table 1. We used parameter setting $((E, m, n)$: i) $(60, 5, 9)$, ii) $(84, 6, 11)$, iii) $(112, 7, 13)$, where $l = 3$ (which are same setting with experiment in Fig 2(d)). We indicate the verification time of HGT, AVCC, and GVC in Table 1. We can see that the verification time of GVC remains the same value, while other methods show an increase in the verification time. This is because other methods require the verification of more results as the recovery threshold increases, but GVC has the same number of groups that needs to be verified. Moreover, to show the benefits of GVC, we conduct additional experiments. We experimented with the same parameter setting in Fig 2(a) when $l = 3$ and only changed the size of the matrix ($m, n = 5, 9$, $E = 60$, $l = 3$ and $a = b = c = N$). We have changed the size of input matrix $N$ from 900 to 3600 to show the benefit of GVC as $N$ increases, and indicate the verification time in Table 1. As we can see in the table, verification gap between GVCC and other scheme becomes larger as $N$ increases. In specific, HGT shows far larger verification time than GVC. Moreover, gap between AVCC and GVC increases $\times 2.66$ when $N$ becomes $\times 2$, and $\times 9.36$ when $N$ becomes $\times 4$.

## 7 CONCLUSION

In this paper, we propose GVCC, a robust and fast distributed computing scheme for matrix multiplication that can deal with two important problems: adversarial attacks and straggler effects. By combining the verifiable computing with coded computing, GVCC allows a master to verify a group of computation results at a time, reducing verification time to find adversarial workers and provide resiliency to straggler effects simultaneously. Consequently, GVCC is shown to have significant reduction in verification and overall processing time, and experimental results demonstrate the effectiveness of GVCC in distributed computing systems.

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
