# OpenReview forum: "Group-wise Verifiable Distributed Computing for Machine Learning under Adversarial Attacks"
_ICLR.cc/2023/Conference — Submitted to ICLR 2023_

### Official Review · Reviewer_d7sM · 2022-10-21

**Confidence:** 2
**Clarity, Quality, Novelty And Reproducibility:** No comments.
**Correctness:** 3
**Technical Novelty And Significance:** 3
**Empirical Novelty And Significance:** 3
**Recommendation:** 6

**Strength And Weaknesses:**

Strength:

1. The motivation is clear.

Weaknesses:

I have no significant comments about this submission since the targeted topic is not relevant to my research area. In fact. I do not know why this paper is assigned to me. Therefore, I provide a score of 6 for this paper.

**Summary Of The Paper:**

This submission proposed a distributed computing scheme for matrix multiplication that can handle two problems: adversarial attacks and straggler effects. The experimental results demonstrate the robustness and fastness of the proposed approach by comparing it with two baselines.

**Summary Of The Review:**

I have no recommendation for this paper. However, I will decide my score based on other reviewers' comments after the authors' response time.

---

### Official Review · Reviewer_4Emk · 2022-10-24

**Confidence:** 3
**Correctness:** 3
**Technical Novelty And Significance:** 2
**Empirical Novelty And Significance:** 2
**Recommendation:** 3

**Clarity, Quality, Novelty And Reproducibility:**

The results of the paper should be reproducible and the description about method is clear, but this paper is not good enough. The experimental section can be greatly improved.

**Strength And Weaknesses:**

(1)According to the experimental results, group-wise verifiable coded computing can indeed improve the efficiency under different settings, however, this paper only describes the experimental settings in general without enough comparative analysis. The perspective to measure the experimental results is single, and the compared objects are not sufficient. It is necessary to add specific experiments for adversarial workers and the effects of stragglers, and try to apply the proposed method on more complex tasks.
(2)The structure of this paper can be adjusted and some of the statements in the paper could be better.
(3)The proposed method is effective and the description in this paper is detailed. However, method description should be less and experimental analysis could be added.
(4)Although the proposed method can alleviate both problems at the same time, the target task is relatively monotonous.

**Summary Of The Paper:**

This paper proposes group-wise verifiable coded computing with coding techniques and group-wise verification to tackle adversarial workers and the effects of stragglers.
The authors demonstrate the performance of group-wise verifiable coded computing in term of overall processing time and verification time.

**Summary Of The Review:**

In a word, this paper is below the average level of other accepted papers. The proposed method is effective, but the experimental part needs to be improved and more innovations need to be explored.

---

### Official Review · Reviewer_Ma5n · 2022-10-24

**Confidence:** 4
**Correctness:** 4
**Technical Novelty And Significance:** 3
**Empirical Novelty And Significance:** Not applicable
**Recommendation:** 8

**Clarity, Quality, Novelty And Reproducibility:**

The proposed design is well presented, and detailed discussions on their connection to prior works are provided.

**Strength And Weaknesses:**

This manuscript was earlier submitted to Neurips 2022, and my main concerns were on the clarity and comparisons of prior works. The authors have revised the paper significantly. As far as the reviewer can see, all major issues on those aspects are resolved. I only have one minor suggestion on the definition of the recovery threshold. In prior works that consider strong adversaries and apply forward error-correcting approaches, the recovery threshold counts the total number of workers that returns the computation result, but here the definition only counts non-adversarial workers. The reviewer understands that in the framework of verifiable computing, the version presented in this work can be expressed in a simpler formula, which could confuse readers who have seen the other version. The reviewer suggests adding a discussion immediately after the definition of recovery threshold to clarify this difference.

**Summary Of The Paper:**

This paper proposed an approach for distributedly multiplying two large matrices while handling active adversaries and stragglers. The design is within the framework of verifiable computing, where the adversaries are detected before recovering the final results. However, by modifying the encoding polynomials (using squeezed polynomial codes particularly), the required number of verifications is reduced. Therefore, the proposed design reduces the overall computation time compared to earlier proposed approaches, although requiring slightly stronger assumptions on the adversaries.

**Summary Of The Review:**

The work in this manuscript is well-presented and interesting. The reviewer only observed one minor issue, which is easily fixable.

---

### Official Review · Reviewer_Z7en · 2022-10-28

**Confidence:** 3
**Correctness:** 3
**Technical Novelty And Significance:** 2
**Empirical Novelty And Significance:** 1
**Recommendation:** 3

**Clarity, Quality, Novelty And Reproducibility:**

- The technical parts are explained quite clearly. The introduction and the paper is written in quite a winding way, and does not state the main contribution directly, I would recommend writing a more focused introduction.
- The weaknesses section of the view above also provides examples of lack of clarity in terms of experiments.
- The statements pertaining to Chebyshev polynomials and their applicability to condition seem flawed. This is because Chebyshev polynomials are defined over reals (and this assumption is also made in the appendix in the paper), but Proposition 1 requires a polynomial over finite fields.


**Strength And Weaknesses:**

Strengths:
+ The simultaneous handling of polynomial-based verification and stragglers in a distributed computing context seems new.

Weaknesses;
- A natural scheme is to verify each node separately to check if the result is adversarial. The benefit of the proposed scheme over this approach, as I understand it, is that it is computationally more efficient - but the gain appears to be small for large matrices. Assuming that all matrices are $N \times N$,  verification is of order $O(N^2)$. The proposed scheme performs one verification for every $m$ nodes, instead of one for every node, but I am not convinced that this is practically very important when $N \gg m$. Note that gain in computational complexity comes at the cost of a significant assumption - that the adversarial nodes do not collude. This is quite a weak assumption in the context of security literature.

- Although matrix multiplication is important, the paper (main body) does not directly test their results in the context of a machine learning application. The supplemental section does seem to have an application, but it is too sparse in the details for reproduciblity (e.g., how are the codes applied? What is the developed accuracy? What are the wallclock times of various schemes?).


**Summary Of The Paper:**

The paper develops techniques for distributed matrix multiplication with adversarially corrupt nodes and stragglers.  A error correction coding scheme is developed that simultaneously detects corrupt computations via random-key-based verification, and can decode in the presence of stragglers.

**Summary Of The Review:**

The paper has several major weaknesses including (1) a weak model for adversary, (2) lack of precision in writing some parts, and (2) lack of satisfactory experimental evaluation details for the ML applications mentioned. The positive side (lower computation overhead) seems too thin for large matrices to compensate for the above drawbacks.

---

### Decision · Program_Chairs · 2023-01-20

**Decision:**

Reject

**Justification For Why Not Higher Score:**

The paper has some technical flaws

**Justification For Why Not Lower Score:**

N/A

**Metareview: Summary, Strengths And Weaknesses:**

In this paper, the authors propose a solution to the problem of adversarial attacks and straggler effects in distributed computing. Their Group-wise Verifiable Coded Computing (GVCC) approach leverages coding techniques and group-wise verification to provide robustness to adversarial attacks and resiliency to straggler effects. The key idea of GVCC is to verify groups of computation results from workers at a time, while providing resilience to stragglers through encoding tasks assigned to workers with group-wise verifiable codes. The authors also provide experimental results that suggest GVCC outperforms existing methods in terms of overall processing time and verification time for executing matrix multiplication.

The reviewers thought the paper is easy to follow and that the technical parts are explained clearly. The reviewers however thought (1) intro is a bit verbose and does not state the main contribution clearly (2) unclear experiments (3) flawed statements w.r.t. Chebyshev polys (4) an a variety of more detailed technical concerns. The authors rebuttal did not address these issues and in particular some of the more technical questions adequately. Therefore I can not recommend acceptance of the paper in its current version.